# Temporal Spike Sequence Learning via Backpropagation for Deep Spiking Neural Networks

**Wenrui Zhang**
University of California, Santa Barbara
Santa Barbara, CA 93106
wenruizhang@ucsb.edu

**Peng Li**
University of California, Santa Barbara
Santa Barbara, CA 93106
lip@ucsb.edu

## Abstract

Spiking neural networks (SNNs) are well suited for spatio-temporal learning and implementations on energy-efficient event-driven neuromorphic processors. However, existing SNN error backpropagation (BP) methods lack proper handling of spiking discontinuities and suffer from low performance compared with the BP methods for traditional artificial neural networks. In addition, a large number of time steps are typically required to achieve decent performance, leading to high latency and rendering spike based computation unscalable to deep architectures. We present a novel Temporal Spike Sequence Learning Backpropagation (TSSL-BP) method for training deep SNNs, which breaks down error backpropagation across two types of inter-neuron and intra-neuron dependencies and leads to improved temporal learning precision. It captures inter-neuron dependencies through presynaptic firing times by considering the all-or-none characteristics of firing activities, and captures intra-neuron dependencies by handling the internal evolution of each neuronal state in time. TSSL-BP efficiently trains deep SNNs within a much shortened temporal window of a few steps while improving the accuracy for various image classification datasets including CIFAR10.

## 1 Introduction

Spiking neural networks (SNNs), a brain-inspired computational model, have gathered significant interests [9]. The spike-based operational principles of SNNs not only allow information coding based on efficient temporal codes and give rise to promising spatiotemporal computing power but also render energy-efficient VLSI neuromorphic chips such as IBM's TrueNorth [1] and Intel's Loihi [6]. Despite the recent progress in SNNs and neuromorphic processor designs, fully leveraging the theoretical computing advantages of SNNs over traditional artificial neural networks (ANNs) [17] to achieve competitive performance in wide ranges of challenging real-world tasks remains difficult.

Inspired by the success of error backpropagation (BP) and its variants in training conventional deep neural networks (DNNs), various SNNs BP methods have emerged, aiming at attaining the same level of performance [4, 16, 24, 13, 21, 28, 11, 2, 3, 29]. Although many appealing results are achieved by these methods, developing SNNs BP training methods that are on a par with the mature BP tools widely available for training ANNs today is a nontrivial problem [22].

Training of SNNs via BP are challenged by two fundamental issues. First, from an algorithmic perspective, the complex neural dynamics in both spatial and temporal domains make the BP process obscure. Moreover, the errors are hard to be precisely backpropagated due to the non-differentiability of discrete spike events. Second, a large number of time steps are typically required for emulating SNNs in time to achieve decent performance, leading to high latency and rendering spike based computation unscalable to deep architectures. It is desirable to demonstrate the success of BP in training deeper SNNs achieving satisfactory performances on more challenging datasets. We propose a

new SNNs BP method, called temporal spike sequence learning via BP (TSSL-BP), to learn any target output temporal spiking sequences. TSSL-BP acts as a universal training method for any employed spike codes (rate, temporal, and combinations thereof). To tackle the above difficulties, TSSL-BP breaks down error backpropagation across two types of inter-neuron and intra-neuron dependencies, leading to improved temporal learning precision. It captures the inter-neuron dependencies within an SNN by considering the all-or-none characteristics of firing activities through presynaptic firing times; it considers the internal evolution of each neuronal state through time, capturing how the intra-neuron dependencies between different firing times of the same presynaptic neuron impact its postsynaptic neuron. The efficacy and precision of TSSL-BP allows it to successfully train SNNs over a very short temporal window, e.g. over 5-10 time steps, enabling ultra-low latency spike computation. As shown in Section 4, TSSL-BP signficantly improves accuracy and runtime efficiency of BP training on several well-known image datasets of MNIST [15], NMNIST [19], FashionMNIST [26], and CIFAR10 [14]. Specifically, it achieves up to $3.98\%$ accuracy improvement over the previously reported SNN work on CIFAR10, a challenging dataset for all prior SNNs BP methods.

## 2  Background

### 2.1  Existing Backpropagation methods for SNNs

One of the earliest SNNs BP methods is the well-known SpikeProp algorithm [4]. However, SpikeProp and its variants [5, 10, 27] are still limited to single spike per output neuron without demonstrating competitive performance on real-world tasks. In addition, [7, 8, 12, 20] train ANNs and then approximately convert them to SNNs. Nevertheless, such conversion leads to approximation errors and cannot exploit SNNs' temporal learning capability.

Recently, training SNNs with BP under a firing rate (or activity level) coded loss function has been shown to deliver competitive performances [16, 24, 21, 13, 28, 11, 2, 29]. Among them, [16] does not consider the temporal correlations of neural activities and treats spiking times as noise to allow error gradient computation. [21, 24, 28, 2] capture the temporal effects by performing backpropagation through time (BPTT) [23]. However, they get around the non-differentiability of spike events by approximating the spiking process using the surrogate gradient method [18]. These approximations lead to inconsistency between the computed gradient and target loss, and thus degrade training performance. [11] presents a BP method for recurrent SNNs based on a novel combination of a gate function and threshold-triggered synaptic model that are introduced to handle non-differentiability of spikes. In this work, depolarization of membrane potential within a narrow active zone below the firing threshold also induces graded postsynaptic current. [13, 29] present spike-train level BP methods by capturing the effects of spiking neurons aggregated at the spike train level. However, the length of spike trains over which BP is applied need to be long enough, leading to long inference latency and high training cost.

[25] can train SNNs over a relatively small number of time steps by adding optimization techniques such as neuron normalization and population decoding. Since its core lies at the method of [24], it still approximates the all-or-none firing characteristics of spiking neurons by a continuous activation function, causing the same problems introduced before.

In this work, we propose the TSSL-BP method as a universal training method for any employed spike codes. It can not only precisely capture the temporal dependencies but also allow ultra-low latency inference and training over only five time steps while achieving excellent accuracies.

### 2.2  Spiking Neuron Model

SNNs employ more biologically plausible spiking neuron models than ANNs. In this work, we adopt the leaky integrate-and-fire (LIF) neuron model and synaptic model [9].

Consider the input spike train from pre-synaptic neuron $j$ to neuron $i$: $s_j(t) = \sum_{t_j^{(f)}} \delta(t - t_j^{(f)})$, where $t_j^{(f)}$ denotes a particular firing time of presynaptic neuron $j$. The incoming spikes are converted into an (unweighted) postsynaptic current (PSC) $a_j(t)$ through a synaptic model. The neuronal

membrane voltage $u_i(t)$ at time $t$ for neuron $i$ is given by

$$\tau_m \frac{du_i(t)}{dt} = -u_i(t) + R \sum_j w_{ij} a_j(t) + \eta_i(t), \tag{1}$$

where $R$ and $\tau_m$ are the effective leaky resistance and time constant of the membrane, $w_{ij}$ is the synaptic weight from pre-synaptic neuron $j$ to neuron $i$, $a_j(t)$ is the (unweighted) postsynaptic potential (PSC) induced by the spikes from pre-synaptic neuron $j$, and $\eta(t)$ denotes the reset function.

The PSC and the reset function can be written as

$$a_j(t) = (\epsilon * s_j)(t), \qquad \eta_i(t) = (\nu * s_i)(t), \tag{2}$$

where $\epsilon(\cdot)$ and $\nu(\cdot)$ are the spike response and reset kernel, respectively. In this work, we adopt a first order synaptic model as the spike response kernel which is expressed as:

$$\tau_s \frac{a_j(t)}{dt} = -a_j(t) + s_j(t), \tag{3}$$

where $\tau_s$ is the synaptic time constant.

The reset kernel reduces the membrane potential by a certain amount $\Delta_R$, where $\Delta_R$ is equal to the firing threshold right after the neuron fires. Considering the discrete time steps simulation, we use the fixed-step first-order Euler method to discretize (1) to

$$u_i[t] = (1 - \frac{1}{\tau_m}) u_i[t-1] + \sum_j w_{ij} a_j[t] + \eta_i[t]. \tag{4}$$

The ratio of R and $\tau_m$ is absorbed into the synaptic weight. The reset function $\eta_i[t]$ represents the firing-and-resetting mechanism of the neuron model. Moreover, the firing output of the neuron is expressed as

$$s_i[t] = H\left(u_i[t] - V_{th}\right), \tag{5}$$

where $V_{th}$ is the firing threshold and $H(\cdot)$ is the Heaviside step function.

## 3 Methods

### 3.1 Forward Pass

Without loss of generality, we consider performing BP across two adjacent layers $l-1$ and $l$ with $N_{l-1}$ and $N_l$ neurons, respectively, in a fully-connected feedforward SNNs as shown in Figure 1. The procedure can be also applied to convolutional and pooling layers. Denote the presynaptic weights by $\boldsymbol{W}^{(l)} = \left[\boldsymbol{w}_1^{(l)}, \cdots, \boldsymbol{w}_{N_l}^{(l)}\right]^T$,
where $\boldsymbol{w}_i^{(l)}$ is a column vector of weights from all the neurons in layer $l-1$ to the neuron $i$ of layer $l$. In addition, we also denote PSCs from neurons in layer $l-1$ by $\boldsymbol{a}^{(l-1)}[t] = \left[a_1^{(l-1)}[t], \cdots, a_{N_{l-1}}^{(l-1)}[t]\right]^T$, spike trains output

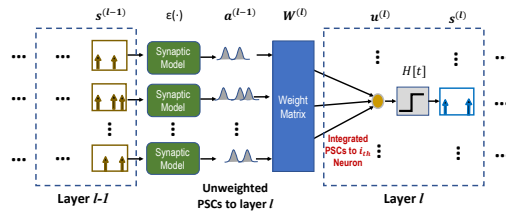

Figure 1: Forward evaluation pass of SNNs.

of the $l-1$ layer by $\boldsymbol{s}^{(l-1)}[t] = \left[s_1^{(l-1)}[t], \cdots, s_{N_{l-1}}^{(l-1)}[t]\right]^T$, membrane potentials and the corresponding output spike trains of the $l$ layer neurons respectively by $\boldsymbol{u}^{(l)}[t] = \left[u_1^{(l)}[t], \cdots, u_{N_l}^{(l)}[t]\right]^T$ and $\boldsymbol{s}^{(l)}[t] = \left[s_1^{(l)}[t], \cdots, s_{N_l}^{(l)}[t]\right]^T$, where variables associated with neurons in the layer $l$ have $l$ as the superscript.

The forward propagation between the two layers is described as

$$\boldsymbol{a}^{(l-1)}[t] = (\epsilon * \boldsymbol{s}^{(l-1)})[t], \quad \boldsymbol{u}^{(l)}[t] = (1 - \frac{1}{\tau_m})\boldsymbol{u}^{(l)}[t-1] + \boldsymbol{W}^{(l)}\boldsymbol{a}^{(l-1)}[t] + (\nu * \boldsymbol{s}^{(l)})[t],$$
$$\boldsymbol{s}^{(l)}[t] = H\left(\boldsymbol{u}^{(l)}[t] - V_{th}\right). \tag{6}$$

In the forward pass, the spike trains $\boldsymbol{s}^{(l-1)}[t]$ of the $l-1$ layer generate the (unweighted) PSCs $\boldsymbol{a}^{(l-1)}[t]$ according to the synaptic model. Then, $\boldsymbol{a}^{(l-1)}[t]$ are multiplied the synaptic weights and passed onto the neurons of layer $l$. The integrated PSCs alter the membrane potentials and trigger the output spikes of the layer $l$ neurons when the membrane potentials exceed the threshold.

## 3.2 The Loss Function

The goal of the proposed TSSL-BP method is to train a given SNN in such a way that each output neuron learns to produce a desired firing sequence arbitrarily specified by the user according to the input class label. Denote the desired and the actual spike trains in the output layer by $\boldsymbol{d} = [\boldsymbol{d}[t_0], \cdots, \boldsymbol{d}[t_{N_t}]]$ and $\boldsymbol{s} = [\boldsymbol{s}[t_0], \cdots, \boldsymbol{s}[t_{N_t}]]$ where $N_t$ is the number of the considered time steps, $\boldsymbol{d}[t]$ and $\boldsymbol{s}[t]$ the desired and actual firing events for all output neurons at time $t$, respectively.

The loss function $L$ can be defined using any suitable distance function measuring the difference between $\boldsymbol{d}$ and $\boldsymbol{s}$. In this work, the loss function is defined by the total square error summed over each output neuron at each time step:

$$L = \sum_{k=0}^{N_t} E[t_k] = \sum_{k=0}^{N_t} \frac{1}{2}((\epsilon * \boldsymbol{d})[t_k] - (\epsilon * \boldsymbol{s})[t_k])^2, \tag{7}$$

where $E[t]$ is the error at time $t$ and $\epsilon(\cdot)$ a kernel function which measures the so-called Van Rossum distance between the actual spike train and desired spike train.

## 3.3 Temporal Spike Sequence Learning via Backpropagation (TSSL-BP) Method

From the loss function (7), we define the error $E[t_k]$ at each time step. $E[t_k]$ is based on the output layer firing spikes at $t_k$ which further depend on all neuron states $\mathbf{u}[t]$, $t \leq t_k$. Below we consider these temporal dependencies and derive the main steps of the proposed TSSL-BP method. Full details of the derivation are provided in Section 1 of the Supplementary Material.

Using (7) and the chain rule, we obtain

$$\frac{\partial L}{\partial \boldsymbol{W}^{(l)}} = \sum_{k=0}^{N_t} \sum_{m=0}^{k} \frac{\partial E[t_k]}{\partial \boldsymbol{u}^{(l)}[t_m]} \frac{\partial \boldsymbol{u}^{(l)}[t_m]}{\partial \boldsymbol{W}^{(l)}} = \sum_{m=0}^{N_t} \boldsymbol{a}^{(l-1)}[t_m] \sum_{k=m}^{N_t} \frac{\partial E[t_k]}{\partial \boldsymbol{u}^{(l)}[t_m]}. \tag{8}$$

Similar to the conventional backpropagation, we use $\delta$ to denote the back propagated error at time $t_m$ as $\boldsymbol{\delta}^{(l)}[t_m] = \sum_{k=m}^{N_t} \frac{\partial E[t_k]}{\partial \boldsymbol{u}^{(l)}[t_m]}$. $\boldsymbol{a}^{(l-1)}[t_m]$ can be easily obtained from (6). $\boldsymbol{\delta}^{(l)}[t_m]$ is considered in two cases.

**[$l$ is the output layer.]** The $\boldsymbol{\delta}^{(l)}[t_m]$ can be computed from

$$\boldsymbol{\delta}^{(l)}[t_m] = \sum_{k=m}^{N_t} \frac{\partial E[t_k]}{\partial \boldsymbol{a}^{(l)}[t_k]} \frac{\partial \boldsymbol{a}^{(l)}[t_k]}{\partial \boldsymbol{u}^{(l)}[t_m]}. \tag{9}$$

The first term of (9) can be obtained directly from the loss function. The second term $\frac{\partial \boldsymbol{a}^{(l)}[t_k]}{\partial \boldsymbol{u}^{(l)}[t_m]}$ is the key part of the TSSL-BP method and is discussed in the following sections.

**[$l$ is a hidden layer.]** $\boldsymbol{\delta}^{(l)}[t_m]$ is derived using the chain rule and (6).

$$\boldsymbol{\delta}^{(l)}[t_m] = \sum_{j=m}^{N_t} \sum_{k=m}^{j} \frac{\partial \boldsymbol{a}^{(l)}[t_k]}{\partial \boldsymbol{u}^{(l)}[t_m]} \left( \frac{\partial \boldsymbol{u}^{(l+1)}[t_k]}{\partial \boldsymbol{a}^{(l)}[t_k]} \frac{\partial E[t_j]}{\partial \boldsymbol{u}^{(l+1)}[t_k]} \right) = (\boldsymbol{W}^{(l+1)})^T \sum_{k=m}^{N_t} \frac{\partial \boldsymbol{a}^{(l)}[t_k]}{\partial \boldsymbol{u}^{(l)}[t_m]} \boldsymbol{\delta}^{(l+1)}[t_k]. \tag{10}$$

(10) maps the error $\boldsymbol{\delta}$ from layer $l+1$ to layer $l$. It is obtained from the fact that membrane potentials $\boldsymbol{u}^{(l)}$ of the neurons in layer $l$ influence their (unweighted) corresponding postsynaptic currents (PSCs) $\boldsymbol{a}^{(l)}$ through fired spikes, and $\boldsymbol{a}^{(l)}$ further affect the membrane potentials $\boldsymbol{u}^{(l+1)}$ in the next layer.

### 3.3.1 Key challenges in SNN BackPropagation

As shown above, for both the output layer and hidden layers, once $\frac{\partial \boldsymbol{a}^{(l)}[t_k]}{\partial \boldsymbol{u}^{(l)}[t_m]} (t_k \geq t_m)$ are known, the error $\boldsymbol{\delta}$ can be back propagated and the gradient of each layer can be calculated.

Importantly, the dependencies of the PSCs on the corresponding membrane potentials of the presynaptic neurons reflected in $\frac{\partial \boldsymbol{a}^{(l)}[t_k]}{\partial \boldsymbol{u}^{(l)}[t_m]}(t_k \geq t_m)$ are due to the following spiking neural behaviors: a change in the membrane potential may bring it up to the firing threshold, and hence activate the corresponding neuron by generating a spike, which in turn produces a PSC. Computing $\frac{\partial \boldsymbol{a}^{(l)}[t_k]}{\partial \boldsymbol{u}^{(l)}[t_m]}(t_k \geq t_m)$ involves the activation of each neuron, i.e. firing a spike due to the membrane potential's crossing the firing threshold from below. Unfortunately, the all-or-none firing characteristics of spiking neurons makes the activation function nondifferentiable, introducing several key challenges.

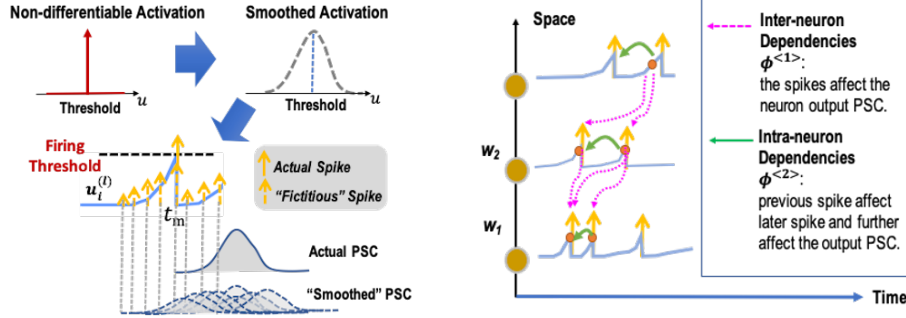

Figure 2: "Fictitious" smoothing of activation  Figure 3: Inter/Intra neuron dependencies.

A typical strategy in dealing with the non-differentiability of the activation is to smooth the activation function by approximating it using a differentiable curve [24] as shown in the Figure 2, or a continuous probability density function [21], which is similar to the former approach in spirit. However, these approaches effectively spread each discrete firing spike continuously over time, converting one actual spike to multiple "fictitious" spikes and also generating multiple "fictitious" PSCs displaced at different time points. We stress that while smoothing circumvents the numerical challenges brought by non-differentiability of the spiking activation, it effectively alters the underlying spiking neuron model and firing times, and leads to degraded accuracy in the error gradient computation. It is important to reflect that spike timing is the hallmark of spiking neural computation, altering firing times in BP can hamper precise learning of the targeted firing sequences as pursued in this paper.

### 3.3.2 The Main Ideas Behind TSSL-BP

TSSL-BP addresses the two key limitations of the prior BP methods: lack proper handling of spiking discontinuities (leading to loss of temporal precision) and need for many time steps (i.e. high latency) to ensure good performance. TSSL-BP computes $\frac{\partial \boldsymbol{a}^{(l)}[t_k]}{\partial \boldsymbol{u}^{(l)}[t_m]}$ across two categories of spatio-temporal dependencies in the network: **inter-neuron** and **intra-neuron** dependencies. As shown in Figure 3, our key observations are: 1) temporal dependencies of a postsynaptic neuron on any of its presynaptic neurons *only* take place via the presynaptic spikes which generate PSCs to the postsynaptic neuron, and shall be considered as inter-neuron dependencies; 2) furthermore, the timing of one presynaptic spike affects the timing of the immediately succeeding spike from the same presynaptic neuron through the intra-neuron temporal dependency. The timing of the first presynaptic spike affects the PSC produced by the second spike, and has additional impact on the postsynaptic neuron through this indirect mechanism. In the following, we show how to derive the $\frac{\partial a_i^{(l)}[t_k]}{\partial u_i^{(l)}[t_m]}$ for each neuron $i$ in layer $l$. We denote $\phi_i^{(l)}(t_k, t_m) = \frac{\partial a_i^{(l)}[t_k]}{\partial u_i^{(l)}[t_m]} = \phi_i^{(l)<1>}(t_k, t_m) + \phi_i^{(l)<2>}(t_k, t_m)$, where $\phi_i^{(l)<1>}(t_k, t_m)$ represents the inter-neuron dependencies and $\phi_i^{(l)<2>}(t_k, t_m)$ is the intra-neuron dependencies.

### 3.3.3 Inter-Neuron Backpropagation

Instead of performing the problematic activation smoothing, we critically note that the all-or-none characteristics of firing behavior is such that a PSC waveform is only triggered at a presynaptic firing time. Specially, as shown in Figure 4, a perturbation $\Delta u_i^{(l)}$ of $u_i^{(l)}[t_m]$, i.e, due to weight updates, may result in an incremental shift in the firing time $\Delta t$, which in turn shifts the onset of the PSC

waveform corresponding to the shifted spike, leading to a perturbation $\Delta a_i^{(l)}$ of $a_i^{(l)}[t_k]$. We consider this as an inter-neuron dependency since the change in PSC ($\Delta a_i^{(l)}$) alters the membrane potential of the postsynaptic neuron in the next layer.

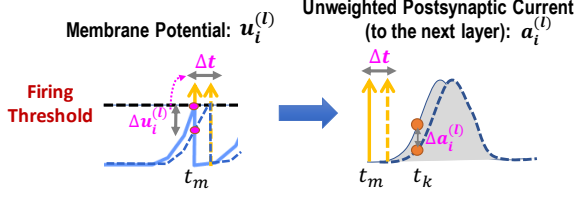

Figure 4: PSC dependencies on presynaptic potential.

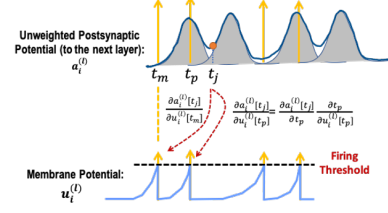

Figure 5: Inter-neuron dependencies.

We make two important points: 1) we shall capture the inter-neuron dependencies via (the incremental changes of) the presynaptic firing times, which precisely corresponds to how different neurons interact with each other in an SNN; and 2) the *inter-neuron* dependency of each neuron $i$'s PSC $a_i^{(l)}$ at $t_k$ on its membrane potential $u_i^{(l)}$ at $t_m$ happens only if the neuron fires at $t_m$. In general, $a_i^{(l)}[t_k]$'s inter-neuron dependencies on *all* preceding firing times shall be considered. Figure 5 shows the situation where $a_i^{(l)}[t_k]$ depends on two presynaptic firing times $t_m$ and $t_p$. Conversely, the inter-neuron dependencies $\phi_i^{(l)<1>}(t_k, t_m) = 0$ if $t_k < t_m$ or there is no spike at $t_m$. Assuming that the presynaptic neuron $i$ spikes at $t_m$, The inter-neuron dependencies is

$$\phi_i^{(l)<1>}(t_k, t_m) = \frac{\partial a_i^{(l)}[t_k]}{\partial t_m} \frac{\partial t_m}{\partial u_i^{(l)}[t_m]}, \tag{11}$$

where, importantly, the chain rule is applied through the presynaptic firing time $t_m$.

From (2), the two parts part of (11) can be calculated as

$$\frac{\partial a_i^{(l)}[t_k]}{\partial t_m} = \frac{\partial (\epsilon * s_i^{(l)}[t_m])[t_k]}{\partial t_m}, \qquad \frac{\partial t_m}{\partial u_i^{(l)}[t_m]} = \frac{-1}{\frac{\partial u_i^{(l)}[t_m]}{\partial t_m}}, \tag{12}$$

where $\frac{\partial u_i^{(l)}[t_m]}{\partial t_m}$ is obtained by differentiating (4).

### 3.3.4 Intra-Neuron Backpropagation

Now we consider the intra-neuron dependency $\phi_i^{(l)<2>}(t_k, t_m)$ defined between an arbitrary time $t_k$ and a presynaptic firing time $t_m$ ($t_m < t_k$). From Section 3.3.3, the presynpatic firing at time $t_m$ invokes a continuous PSC which has a direct impact on the postsynaptic potential at time $t_k$, which is an inter-neuron dependency. On the other hand, $\phi_i^{(l)<2>}(t_k, t_m)$ corresponds to an indirect effect on the postsynaptic potential via the intra-neuron dependency.

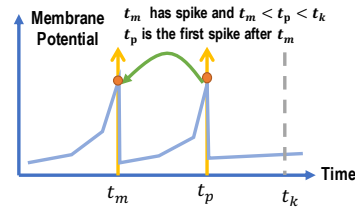

Figure 6: Intra-neuron dependencies.

We consider $\phi_i^{(l)<2>}(t_k, t_m)$ specifically under the context of the adopted LIF model, which may occur if the presynaptic neuron spikes at $t_p$ immediately following $t_m$ such that $t_m < t_p < t_k$. In this case, the presynatpic membrane potential at $t_m$ not only contributes to a PSC due to the neuron firing, but also affects the membrane potential at the next spike time $t_p$ resulted from the reset occurring at $t_m$ as described in (4). The PSC $a_i^{(l)}[t_k]$ has an inter-neuron dependency on membrane potential $u_i^{(l)}[t_p]$ while $u_i^{(l)}[t_p]$ is further affected by the immediately preceding firing time $t_m$ due to the reset of the presynaptic potential at $t_m$. Recall $s_i^{(l)}[t] = 1$ if neuron $i$ fires at $t$ as in (5). More precisely, $\phi_i^{(l)<2>}(t_k, t_m)$ takes this indirect intra-neuron effect on $a_i^{(l)}[t_k]$ into consideration if

$\exists t_p \in (t_m, t_k)$ such that $s_i^{(l)}[t_p] = 1$ and $s_i^{(l)}[t] = 0 \; \forall t \in (t_m, t_p)$, i.e. no other presynaptic spike exists between $t_m$ and $t_p$

$$\phi_i^{(l)<2>}(t_k, t_m) = \frac{\partial a_i^{(l)}[t_k]}{\partial u_i^{(l)}[t_p]} \frac{\partial u_i^{(l)}[t_p]}{\partial t_m} \frac{\partial t_m}{\partial u_i^{(l)}[t_m]} = \phi_i^{(l)}(t_k, t_p) \frac{\partial(\nu * s_i^{(l)}[t_m])[t_p]}{\partial t_m} \frac{\partial t_m}{\partial u_i^{(l)}[t_m]}, \quad (13)$$

where $\nu(\cdot)$ is the reset kernel and $\frac{\partial t_m}{\partial u_i^{(l)}[t_m]}$ is evaluated by (12). In (13), $\phi_i^{(l)}(t_k, t_p)$ would have been already computed during the backpropagation process since $t_p$ is a presynaptic firing time after $t_m$. Putting the inter-neuron and intra-neuron dependencies together, one of the key derivatives required in the BP process $\phi_i^{(l)}(t_k, t_m) = \frac{\partial a_i^{(l)}[t_k]}{\partial u_i^{(l)}[t_m]}$ with $t_m < t_p < t_k$ is given by

$$\phi_i^{(l)}(t_k, t_m) = \begin{cases} 0 & s_i^{(l)}[t_m] = 0, s_i^{(l)}[t_p] = 0 \; \forall t_p \in (t_m, t_k), \\ \frac{\partial a_i^{(l)}[t_k]}{\partial t_m} \frac{\partial t_m}{\partial u_i^{(l)}[t_m]} & s_i^{(l)}[t_m] = 1, s_i^{(l)}[t_p] = 0 \; \forall t_p \in (t_m, t_k), \\ \frac{\partial a_i^{(l)}[t_k]}{\partial t_m} \frac{\partial t_m}{\partial u_i^{(l)}[t_m]} + \phi_i^{(l)<2>}(t_k, t_m) & s_i^{(l)}[t_m] = 1, s_i^{(l)}[t_p] = 1, s_i^{(l)}[t] = 0 \; \forall t \in (t_m, t_p), \end{cases} \quad (14)$$

where $t_p$ is an arbitrary time between $t_m$ and $t_k$, and $\phi_i^{(l)<2>}(t_k, t_m)$ of (13) is considered.

The complete derivation of TSSL-BP is provided in Section 1 of the Supplementary Material. There are two key distinctions setting our approach apart from the aforementioned activation smoothing. First, the inter-neuron dependencies are only considered at pre-synaptic firing times as opposed to all prior time points, latter of which is the case when the activation smoothing is applied with BPTT. The handling adopted in TSSL-BP is a manifestation of the all-or-none firing characteristics of spiking neurons. Second, as in Figure 4, the key step in backpropagation is the consideration of the incremental change of spiking times, which is not considered in recent SNNs BP works.

## 4 Experiments and Results

We test the proposed TSSL-BP method on four image datasets MNIST [15], N-MNIST [19], Fashion-MNIST [26] and CIFAR10 [14]. We compare TSSL-BP with several state-of-the-art results with the same or similar network sizes including different SNNs BP methods, converted SNNs, and traditional ANNs. The details of practical simulation issues, experimental settings, and datasets preprocessing methods are described in Section 2 of the supplementary material. We have made our Pytorch implementation available online[1]. We expect this work would help move the community forward towards enabling high-performance spiking neural networks simulation within short latency.

### 4.1 MNIST

On MNIST [15], we compares the accuracies of the spiking CNNs trained by the TSSL-BP method with ones trained by other algorithms in Table 1. In our method, the pixel intensities of the image are converted into real-valued spike current to the inputs within a short time window. The proposed TSSL-BP delivers up to $99.53\%$ accuracy and outperforms all other methods except for the ST-RSBP [29] whose accuracy is slightly higher by $0.09\%$. However, compared to ST-RSBP, TSSL-BP can train high-performance SNNs with only 5 time steps, achieving $80\times$ reduction of step count (latency). The accuracy of [29] drops below that of TSSL-BP noticeably under short time windows. In addition, no data augmentation is applied in this experiment, which is adopted in [13] and [29].

### 4.2 N-MNIST

We test the proposed method on N-MNIST dataset [19], the neuromorphic version of the MNIST. The inputs to the networks are spikes rather than real value currents. Table 2 compares the results obtained by different models on N-MNIST. The SNN trained by our proposed approach naturally processes spatio-temporal spike patterns, achieving the start-of-the-art accuracy of $99.40\%$. It is important to note that our proposed method with the accuracy of $99.28\%$ outperforms the best previously reported results in [21], obtaining 10 times fewer time steps which leads to significant latency reduction.

Table 1: Performances of Spiking CNNs on MNIST.

| Methods | Network | Time steps | Epochs | Mean | Stddev | Best |
|---|---|---|---|---|---|---|
| Spiking CNN [16] | 20C5-P2-50C5-P2-200 | > 200 | 150 | | | 99.31% |
| STBP [24] | 15C5-P2-40C5-P2-300 | > 100 | 200 | | | 99.42% |
| SLAYER [21] | 12C5-p2-64C5-p2 | 300 | 100 | 99.36% | 0.05% | 99.41% |
| HM2BP [13] | 15C5-P2-40C5-P2-300 | 400 | 100 | 99.42% | 0.11% | 99.49% |
| ST-RSBP [29] | 15C5-P2-40C5-P2-300 | 400 | 100 | 99.57% | 0.04% | 99.62% |
| This work | 15C5-P2-40C5-P2-300 | **5** | 100 | 99.50% | 0.02% | 99.53% |

20C5: convolution layer with 20 of the $5 \times 5$ filters. P2: pooling layer with $2 \times 2$ filters.

Table 2: Performances on N-MNIST.

| Methods | Network | Time steps | Mean | Stddev | Best | Steps reduction |
|---|---|---|---|---|---|---|
| HM2BP [13] | $400 - 400$ | 600 | 98.88% | 0.02% | 98.88% | 20x |
| SLAYER [21] | $500 - 500$ | 300 | 98.89% | 0.06% | 98.95% | 10x |
| SLAYER [21] | 12C5-P2-64C5-P2 | 300 | 99.20% | 0.02% | 99.22% | 10x |
| This work | 12C5-P2-64C5-P2 | 100 | **99.35**% | 0.03% | **99.40**% | 3.3x |
| This work | 12C5-P2-64C5-P2 | **30** | 99.23% | 0.05% | 99.28% | |

All the experiments in this table train the N-MNIST for 100 epochs

## 4.3 FashionMNIST

We compare several trained fully-connected feedforward SNNs and spiking CNNs on FashionM-NIST [26], a more challenging dataset than MNIST. In Table 3, TSSL-BP achieves 89.80% test accuracy on the fully-connected feedforward network of two hidden layers with each having 400 neurons, outperforming the HM2BP method of [13], which is the best previously reported algorithm for feedforward SNNs. TSSL-BP can also deliver the best test accuracy with much fewer time steps. Moreover, TSSL-BP achieves **92.83**% on the spiking CNN networks, noticeably outperforming the same size non-spiking CNN trained by a standard BP method.

Table 3: Performances on FashionMNIST.

| Methods | Network | Time steps | Epochs | Mean | Stddev | Best |
|---|---|---|---|---|---|---|
| ANN [29] | $400 - 400$ | | 100 | | | 89.01% |
| HM2BP [29] | $400 - 400$ | 400 | 100 | | | 88.99% |
| This work | $400 - 400$ | **5** | 100 | 89.75% | 0.03% | **89.80**% |
| ANN [26] | 32C5-P2-64C5-P2-1024 | | 100 | | | 91.60% |
| This work | 32C5-P2-64C5-P2-1024 | 5 | 100 | 92.69% | 0.09% | **92.83**% |

## 4.4 CIFAR10

Furthermore, we apply the proposed method on the more challenging dataset of CIFAR10 [14]. As shown in Table 4, our TSSL-BP method achieves 89.22% accuracy with a mean of 88.98% and a standard deviation of 0.27% under five trials on the first CNN and achieves 91.41% accuracy on the second CNN architecture. TSSL-BP delivers the best result among a previously reported ANN, SNNs converted from pre-trained ANNs, and the spiking CNNs trained by the STBP method of [25]. CIFAR10 is a challenging dataset for most of the existing SNNs BP methods since the long latency required by those methods makes them hard to scale to deeper networks. The proposed TSSL-BP not only achieves up to 3.98% accuracy improvement over the work of [25] without the additional optimization techniques including neuron normalization and population decoding which are employed in [25], but also utilizes fewer time steps.

Table 4: Performances of CNNs on CIFAR10.

| Methods | Network | Time steps | Epochs | Accuracy |
|---------|---------|------------|--------|----------|
| Converted SNN [12] | CNN 1 | 80 | | 83.52% |
| STBP [25] | CNN 1 | 8 | 150 | 85.24% |
| This work | CNN 1 | **5** | 150 | **89.22**% |
| ANN [25] | CNN 2 | | | 90.49% |
| Converted SNN [20] | CNN 2 | | 200 | 87.46% |
| STBP (without NeuNorm) [25] | CNN 2 | 8 | 150 | 89.83% |
| STBP (with NeuNorm) [25] | CNN 2 | 8 | 150 | 90.53% |
| This work | CNN 2 | **5** | 150 | **91.41**% |

CNN 1: 96C3-256C3-P2-384C3-P2-384C3-256C3-1024-1024
CNN 2: 128C3-256C3-P2-512C3-P2-1024C3-512C3-1024-512

## 4.5 Firing Sparsity

As presented, the proposed TSSL-BP method can train SNNs with low latency. In the meanwhile, the firing activities in well-trained networks also tend to be sparse. To demonstrate firing sparsity, we select two well-trained SNNs, one for the CIFAR10 and the other for the N-MNIST.

The CIFAR10 network is simulated over $5$ time steps. We count the percentage of neurons that fire $0, 1, \ldots, 5$ times, respectively, and average the percentages over $100$ testing samples. As shown in Figure 7, the network's firing activity is sparse. More than $84\%$ of neurons are silent while $12\%$ of neurons fire more than once, and about $4\%$ of neurons fire at every time step.

The N-MNIST network demonstrated here is simulated over $100$ time steps. The firing rate of each neuron is logged. The number of neurons with a certain range of firing rates is counted and averaged over $100$ testing samples. Similarly, as shown in Figure 8, the firing events of the N-MNIST network are also sparse and more than $75\%$ of neurons keep silent. In the meanwhile, there are about $5\%$ of neurons with a firing rate of greater than $10\%$.

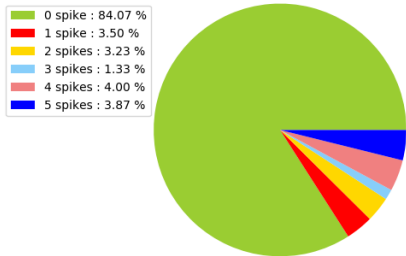
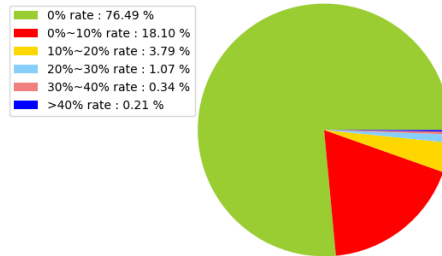

Figure 7: Firing activity on CIFAR10        Figure 8: Firing activity on N-MNIST

## 5 Conclusion

We have presented the novel temporal spike sequence learning via a backpropagation (TSSL-BP) method to train deep SNNs. Unlike all prior SNNs BP methods, TSSL-BP improves temporal learning precision by circumventing the non-differentiability of the spiking activation function while faithfully reflecting the all-or-none firing characteristics and the underlying structure in the dependencies of spiking neural activities in both space and time.

TSSL-BP provides a universal BP tool for learning arbitrarily specified target firing sequences with high accuracy while achieving low temporal latency. This is in contrast with most of the existing SNN BP methods which require hundreds of time steps for achieving decent accuracy. The ability in training and inference over a few time steps results in significant reductions of the computational cost required for training large/deep SNNs, and the decision time and energy dissipation of the SNN model when deployed on either a general-purpose or a dedicated neurormorphic computing platform.

## Broader Impact

Our proposed Temporal Spike Sequence Learning Backpropagation (TSSL-BP) method is able to train deep SNNs while achieving the state-of-the-art efficiency and accuracy. TSSL-BP breaks down error backpropagation across two types of inter-neuron and intra-neuron dependencies and leads to improved temporal learning precision. It captures inter-neuron dependencies through presynaptic firing times by considering the all-or-none characteristics of firing activities, and captures intra-neuron dependencies by handling the internal evolution of each neuronal state in time.

Spiking neural networks offer a very appealing biologically plausible model of computation and may give rise to ultra-low power inference and training on recently emerged large-scale neuromorphic computing hardware. Due to the difficulties in dealing with the all-or-one characteristics of spiking neurons, however, training of SNNs is a major present challenge and has limited wide adoption of SNNs models.

The potential impacts of this work are several-fold:

**1) Precision**: TSSL-BP offers superior precision in learning arbitrarily specified target temporal sequences, outperforming all recently developed the-state-of-the-art SNN BP methods.

**2) Low latency**: TSSL-BP delivers high-precision training over a very short temporal window of a few time steps. This is contrast with many BP methods that require hundreds of time steps for maintaining a decent accuracy. Low latency computation immediately corresponds to fast decision making.

**3) Scalability and energy efficiency**: The training of SNNs is signficantly more costly than that of the conventional neural networks. The training cost is one major bottleneck to training large/deep SNNs in order to achieve competitive performance. The low latency training capability of TSSL-BP reduces the training cost by more than one order of magnitude and also cuts down the energy dissipation of the training and inference on the deployed computing hardware.

**4) Community impact**: TSSL-BP has been prototyped based on the widely adopted Pytorch framework and will be made available to the public. We believe our TSSL-BP code will benefit the brain-inspired computing community from both an algorithmic and neuromorphic computing hardware development perspective.

## Acknowledgments

This material is based upon work supported by the National Science Foundation (NSF) under Grants No.1948201 and No.1940761. Any opinions, findings, conclusions or recommendations expressed in this material are those of the authors and do not necessarily reflect the views of NSF, UC Santa Barbara, and their contractors.

## Footnotes

[1] https://github.com/stonezwr/TSSL-BP

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
