[Supplementary Material]

# Supplementary Materials for: Temporal Spike Sequence Learning via Backpropagation for Deep Spiking Neural Networks

**Wenrui Zhang**
University of California, Santa Barbara
Santa Barbara, CA 93106
wenruizhang@ucsb.edu

**Peng Li**
University of California, Santa Barbara
Santa Barbara, CA 93106
lip@ucsb.edu

## 1 Full Derivation of TSSL-BP

### 1.1 Forward Pass

As shown in Figure 1 of the main manuscript, we denote the presynaptic weights by $\boldsymbol{W}^{(l)} = \left[\boldsymbol{w}_1^{(l)}, \cdots, \boldsymbol{w}_{N_l}^{(l)}\right]^T$, PSCs from neurons in layer $l-1$ by $\boldsymbol{a}^{(l-1)}[t] = \left[a_1^{(l-1)}[t], \cdots, a_{N_{l-1}}^{(l-1)}[t]\right]$, spike trains output of the $l-1$ layer by $\boldsymbol{s}^{(l-1)}[t] = \left[s_1^{(l-1)}[t], \cdots, s_{N_{l-1}}^{(l-1)}[t]\right]$, membrane potentials and the corresponding output spike trains of the $l$ layer neurons respectively by $\boldsymbol{u}^{(l)}[t] = \left[u_1^{(l)}[t], \cdots, u_{N_l}^{(l)}[t]\right]$ and $\boldsymbol{s}^{(l)}[t] = \left[s_1^{(l)}[t], \cdots, s_{N_l}^{(l)}[t]\right]$, where variables associated with neurons in the layer $l$ have $l$ as the superscript.

The forward propagation between the two layers is described as

$$
\begin{aligned}
\boldsymbol{a}^{(l-1)}[t] &= (\epsilon * \boldsymbol{s}^{(l-1)})[t] \\
\boldsymbol{u}^{(l)}[t] &= \theta_\tau \boldsymbol{u}^{(l)}[t-1] + \boldsymbol{W}^{(l)}\boldsymbol{a}^{(l-1)}[t] + (\nu * \boldsymbol{s}^{(l)})[t] \\
\boldsymbol{s}^{(l)}[t] &= H\left(\boldsymbol{u}^{(l)}[t] - V_{th}\right).
\end{aligned}
\tag{1}
$$

### 1.2 The Loss Function

The desired and the actual spike trains in the output layer are denoted by $\boldsymbol{d} = [\boldsymbol{d}[t_0], \cdots, \boldsymbol{d}[t_{N_t}]]$ and $\boldsymbol{s} = [\boldsymbol{s}[t_0], \cdots, \boldsymbol{s}[t_{N_t}]]$ where $N_t$ is the number of the considered time steps, $\boldsymbol{d}[t]$ and $\boldsymbol{s}[t]$ the desired and actual firing events for all output neurons at time $t$, respectively.

In our experiments, the loss function $L$ is defined by the square error for each output neuron at each time step:

$$
L = \sum_{k=0}^{N_t} E[t_k] = \frac{1}{2}\sum_{k=0}^{N_t}(\boldsymbol{d}[t_k] - \boldsymbol{s}[t_k])^2 = \frac{1}{2}||\boldsymbol{d} - \boldsymbol{s}||_2^2,
\tag{2}
$$

where $E[t]$ is the error at time $t$. The loss can be also defined by using a kernel function [10]. In our experiments, we use the spike response kernel $\epsilon(\cdot)$, defining the error at each time step as

$$
E[t] = \frac{1}{2}((\epsilon * \boldsymbol{d})[t] - (\epsilon * \boldsymbol{s})[t])^2 = \frac{1}{2}(\boldsymbol{a_d}[t] - \boldsymbol{a_s}[t])^2.
\tag{3}
$$

## 1.3 Temporal Spike Sequence Learning via Backpropagation (TSSL-BP) Method

We adopt (3) to define the total loss

$$L = \sum_{k=0}^{N_t} E[t_k] = \frac{1}{2} \sum_{k=0}^{N_t} (\boldsymbol{a_d}[t_k] - \boldsymbol{a_s}[t_k])^2. \tag{4}$$

For the neurons in layer $l$, the error gradient with respect to the presynaptic weights matrix $\boldsymbol{W}^{(l)}$ is

$$\frac{\partial L}{\partial \boldsymbol{W}^{(l)}} = \sum_{k=0}^{N_t} \frac{\partial E[t_k]}{\partial \boldsymbol{W}^{(l)}}. \tag{5}$$

(1) reveals that the values of $\boldsymbol{u}^{(l)}$ at time $t_k$ have contribution to all future fires and losses. Using the chain rule, we get

$$\frac{\partial L}{\partial \boldsymbol{W}^{(l)}} = \sum_{k=0}^{N_t} \sum_{m=0}^{k} \frac{\partial E[t_k]}{\partial \boldsymbol{u}^{(l)}[t_m]} \frac{\partial \boldsymbol{u}^{(l)}[t_m]}{\partial \boldsymbol{W}^{(l)}}. \tag{6}$$

By changing the order of summation, (6) can be written as

$$\frac{\partial L}{\partial \boldsymbol{W}^{(l)}} = \sum_{m=0}^{N_t} \frac{\partial \boldsymbol{u}^{(l)}[t_m]}{\partial \boldsymbol{W}^{(l)}} \sum_{k=m}^{N_t} \frac{\partial E[t_k]}{\partial \boldsymbol{u}^{(l)}[t_m]} = \sum_{m=0}^{N_t} \boldsymbol{a}^{(l-1)}[t_m] \sum_{k=m}^{N_t} \frac{\partial E[t_k]}{\partial \boldsymbol{u}^{(l)}[t_m]}. \tag{7}$$

We use $\delta$ to denote the back propagated error at time $t_m$ as $\boldsymbol{\delta}^{(l)}[t_m] = \sum_{k=m}^{N_t} \frac{\partial E[t_k]}{\partial \boldsymbol{u}^{(l)}[t_m]}$.

Therefore, the weights update formula (6) can be written as

$$\frac{\partial L}{\partial \boldsymbol{W}^{(l)}} = \sum_{m=0}^{N_t} \boldsymbol{a}^{(l-1)}[t_m] \boldsymbol{\delta}^{(l)}[t_m]. \tag{8}$$

$\boldsymbol{a}^{(l-1)}[t_m]$ is analogous to the pre-activation in the traditional ANNs which can be easily obtained from (1) in the forward pass. $\boldsymbol{\delta}^{(l)}[t_m]$ is considered in two cases.

[$l$ **is the output layer.**] The $\boldsymbol{\delta}^{(l)}[t_m]$ can be computed from

$$\boldsymbol{\delta}^{(l)}[t_m] = \sum_{k=m}^{N_t} \frac{\partial E[t_k]}{\partial \boldsymbol{a}^{(l)}[t_k]} \frac{\partial \boldsymbol{a}^{(l)}[t_k]}{\partial \boldsymbol{u}^{(l)}[t_m]}. \tag{9}$$

From (4), the first term of (9) is given by

$$\frac{\partial E[t_k]}{\partial \boldsymbol{a}^{(l)}[t_k]} = \frac{1}{2} \frac{\partial (\boldsymbol{a_d}[t_k] - \boldsymbol{a}^{(l)}[t_k])^2}{\partial \boldsymbol{a}^{(l)}[t_k]} = \boldsymbol{a}^{(l)}[t_k] - \boldsymbol{a_d}[t_k]. \tag{10}$$

[$l$ **is a hidden layer.**] $\boldsymbol{\delta}^{(l)}[t_m]$ is derived using the chain rule and (1).

$$\boldsymbol{\delta}^{(l)}[t_m] = \sum_{j=m}^{N_t} \sum_{k=m}^{j} \frac{\partial \boldsymbol{a}^{(l)}[t_k]}{\partial \boldsymbol{u}^{(l)}[t_m]} \left( \frac{\partial \boldsymbol{u}^{(l+1)}[t_k]}{\partial \boldsymbol{a}^{(l)}[t_k]} \frac{\partial E[t_j]}{\partial \boldsymbol{u}^{(l+1)}[t_k]} \right). \tag{11}$$

It is obtained from the fact that membrane potentials $\boldsymbol{u}^{(l)}$ of the neurons in layer $l$ influence their (unweighted) corresponding postsynaptic currents (PSCs) $\boldsymbol{a}^{(l)}$ through fired spikes, and $\boldsymbol{a}^{(l)}$ further affect the membrane potentials $\boldsymbol{u}^{(l+1)}$ in the next layer. By changing the order of summation, maps

the error $\boldsymbol{\delta}$ from layer $l+1$ to layer $l$.

$$
\begin{aligned}
\boldsymbol{\delta}^{(l)}[t_m] &= \sum_{k=m}^{N_t} \frac{\partial \boldsymbol{a}^{(l)}[t_k]}{\partial \boldsymbol{u}^{(l)}[t_m]} \sum_{j=k}^{N_t} \frac{\partial \boldsymbol{u}^{(l+1)}[t_k]}{\partial \boldsymbol{a}^{(l)}[t_k]} \frac{\partial E[t_j]}{\partial \boldsymbol{u}^{(l+1)}[t_k]} \\
&= \sum_{j=m}^{N_t} \frac{\partial \boldsymbol{a}^{(l)}[t_j]}{\partial \boldsymbol{u}^{(l)}[t_m]} \sum_{k=j}^{N_t} \boldsymbol{W}^{(l+1)} \frac{\partial E[t_k]}{\partial \boldsymbol{u}^{(l+1)}[t_j]} \\
&= \sum_{k=m}^{N_t} \frac{\partial \boldsymbol{a}^{(l)}[t_k]}{\partial \boldsymbol{u}^{(l)}[t_m]} (\boldsymbol{W}^{(l+1)})^T \boldsymbol{\delta}^{(l+1)}[t_k] \\
&= (\boldsymbol{W}^{(l+1)})^T \sum_{k=m}^{N_t} \frac{\partial \boldsymbol{a}^{(l)}[t_k]}{\partial \boldsymbol{u}^{(l)}[t_m]} \boldsymbol{\delta}^{(l+1)}[t_k].
\end{aligned}
\tag{12}
$$

The details of the key term $\frac{\partial \boldsymbol{a}^{(l)}[t_k]}{\partial \boldsymbol{u}^{(l)}[t_m]}$ is discussed in Section 3.3.3 and 3.3.4 of the main manuscript. We'll also summarize the derivatives below.

For the key derivative $\frac{\partial a_i^{(l)}[t_k]}{\partial u_i^{(l)}[t_m]}$ $(t_k \geq t_m)$ of each neuron $i$ in layer $l$, we denote $\phi_i^{(l)}(t_k, t_m) = \frac{\partial a_i^{(l)}[t_k]}{\partial u_i^{(l)}[t_m]} = \phi_i^{(l)<1>}(t_k, t_m) + \phi_i^{(l)<2>}(t_k, t_m)$, where $\phi_i^{<1>}(t_k, t_m)$ represents the inter-neuron dependency and $\phi_i^{(l)<2>}(t_k, t_m)$ is the intra-neuron dependency.

Assuming that the presynaptic neuron $i$ spikes at $t_m$, the inter-neuron dependencies can be represented by

$$
\phi_i^{(l)<1>}(t_k, t_m) = \frac{\partial a_i^{(l)}[t_k]}{\partial t_m} \frac{\partial t_m}{\partial u_i^{(l)}[t_m]}.
\tag{13}
$$

From (2) of the main manuscript, the first part of (13) can be calculated as

$$
\frac{\partial a_i^{(l)}[t_k]}{\partial t_m} = \frac{\partial (\epsilon * s_i^{(l)}[t_m])[t_k]}{\partial t_m}.
\tag{14}
$$

We adopt the approach in [1, 3] to compute the second part of (11) as

$$
\frac{\partial t_m}{\partial u_i^{(l)}[t_m]} = \frac{-1}{\frac{\partial u_i^{(l)}[t_m]}{\partial t_m}},
\tag{15}
$$

where $\frac{\partial u_i^{(l)}[t_m]}{\partial t_m}$ is obtained by differentiating (4). In fact, (15) can be precisely derived on certain conditions.

According to the Figure 6 of the main manuscript, in the LIF model, the intra-neuron dependencies is caused by the firing-and-resetting mechanism. More precisely, $\phi_i^{(l)<2>}(t_k, t_m)$ takes this indirect intra-neuron effect on $a_i^{(l)}[t_k]$ into consideration if $\exists t_p \in (t_m, t_k)$ such that $s_i^{(l)}[t_p] = 1$ and $s_i^{(l)}[t] = 0$ $\forall t \in (t_m, t_p)$, i.e. no other presynaptic spike exists between $t_m$ and $t_k$

$$
\begin{aligned}
\phi_i^{<2>}(t_k, t_m) &= \frac{\partial a_i^{(l)}[t_k]}{\partial u_i^{(l)}[t_p]} \frac{\partial u_i^{(l)}[t_p]}{\partial t_m} \frac{\partial t_m}{\partial u_i^{(l)}[t_m]} \\
&= \phi_i(t_k, t_p) \frac{\partial (\nu * s_i^{(l)}[t_m])[t_p]}{\partial t_m} \frac{\partial t_m}{\partial u_i^{(l)}[t_m]},
\end{aligned}
\tag{16}
$$

where $\nu(\cdot)$ is the reset kernel and $\frac{\partial t_m}{\partial u_i^{(l)}[t_m]}$ is evaluated by (15). In (16), $\phi_i^{(l)}(t_k, t_p)$ would have been already computed during the backpropagation process since $t_p$ is a presynaptic firing time after $t_m$.

To sum it up, we obtain the derivative of loss with respect to weight according to TSSL-BP method as follows:

$$\frac{\partial L}{\partial \boldsymbol{W}^{(l)}} = \sum_{m=0}^{N_t} \boldsymbol{a}^{(l-1)}[t_m]\boldsymbol{\delta}^{(l)}[t_m],$$

$$\boldsymbol{\delta}^{(l)}[t_m] = \begin{cases} \sum_{k=m}^{N_t} (\boldsymbol{a}^{(l)}[t_k] - \boldsymbol{a_d}[t_k])\phi_i^{(l)}(t_k, t_m) & \text{for output layer,} \\ (\boldsymbol{W}^{(l+1)})^T \sum_{k=m}^{N_t} \phi_i^{(l)}(t_k, t_m)\boldsymbol{\delta}^{(l+1)}[t_k] & \text{for hidden layers,} \end{cases}$$

$$\phi_i^{(l)}(t_k, t_m) = \begin{cases} 0 & s_i^{(l)}[t_m] = 0, s_i^{(l)}[t_p] = 0 \ \forall t_p \in (t_m, t_k), \\ \frac{\partial a_i^{(l)}[t_k]}{\partial t_m}\frac{\partial t_m}{\partial u_i^{(l)}[t_m]} & s_i^{(l)}[t_m] = 1, s_i^{(l)}[t_p] = 0 \ \forall t_p \in (t_m, t_k), \\ \frac{\partial a_i^{(l)}[t_k]}{\partial t_m}\frac{\partial t_m}{\partial u_i^{(l)}[t_m]} + \phi_i^{(l)<2>}(t_k, t_m) & s_i^{(l)}[t_m] = 1, \exists t_p \text{ such that } s_i^{(l)}[t_p] = 1, s_i^{(l)}[t] = 0 \ \forall t \in (t_m, t_p), \end{cases}$$

(17)

where $s_i^{(l)}[t]$ is the firing function described in (5) of the main manuscript and $t_p$ is an arbitrary time between $t_m$ and $t_k$.

## 2 Experiments and Results

### 2.1 Experimental Settings

All reported experiments are conducted on an NVIDIA Titan XP GPU. The implementation of TSSL-BP is on the Pytorch framework [9]. The experimented SNNs are based on the LIF model described in (4) of the main manuscript. The simulation step size is set to 1 ms. Only a few time steps are used to demonstrate low-latency spiking neural computation. The parameters like thresholds and learning rates are empirically tuned. Table 1 lists the typical constant values adopted in our experiments. For the time constant, we vary the membrane time constant from 2ms to 16ms. The same performance level has been observed. This indicates that the proposed method can train SNNs with dynamical behaviors across different timescales and the empirically observed results are not very sensitive to the choice of membrane time constant. No axon and synaptic delay or refractory period is used nor is normalization. Dropout is only applied for the experiments on CIFAR10. Adam [4] is adopted as the optimizer. The network models we train or compare with are either fully connected feedforward networks or convolutional neural networks (CNNs). The mean and standard deviation (stddev) of the accuracy reported are obtained by repeating the experiments five times.

Table 1: Parameters settings.

| Parameter | Value | Parameter | Value |
| --- | --- | --- | --- |
| simulation step size | 1 ms | Learning Rate $\eta$ | 0.005 |
| Time Constant of Membrane Voltage $\tau_m$ | 4 ms | Time Constant of Synapse $\tau_s$ | 2 ms |
| Threshold $V_{th}$ | 1 mV | Batch Size | 50 |

### 2.2 Input Encoding

For non-spiking datasets such as static images, the most common preprocessing is to use rate coding to convert static real-valued inputs to spiking inputs. However, this requires many time steps for coding the inputs to guarantee good performance. For static images, we directly convert the raw pixel densities into real-valued spike current inputs within a short time window. While for the neuromorphic datasets that originally contain spikes, we still use the spikes as inputs in our experiments.

### 2.3 Handling of Practical Issues

Two practical circumstances need to be taken into consideration as for other spike-time based BP methods like SpikeProp [1, 2]. First, when a spike is produced by the membrane potential $u[t]$ that barely reaches the threshold, the derivative of $u[t]$ w.r.t time is very small. Numerically, this can make (15) large and result in an undesirable large weight update. To mitigate, we set a bound for

this derivative. Second, absence of firing activities in spiking neurons due to low initial weight values block backpropagation through these neurons. We use a warm-up mechanism to bring up the firing activity of the network before applying the BP method. In the warm-up mechanism, we set a threshold for the average firing activity of each layer. If the observed activity is higher than the threshold, TSSL-BP is applied directly. Otherwise, at very low firing activity levels, warm-up is applied first, which uses the continuous sigmoid function of membrane potential to approximate the activation function so that the error can be propagated back even when there is no spike. In practice, the performance of the trained network has no significant dependence on the way in which warm-up is adopted and one epoch training based warm-up is sufficient.

## 2.4 Selection of the Desired Output Spike Patterns

The desired output spike trains (labels) for different classes are manually selected without much optimization effort. In the experiments with 5 time steps, we set two fixed sequences [0, 1, 0, 1, 1] and [0, 0, 0, 0, 0] where 1 represents a spike and 0 means no spike at a given time step. We adopted a simple scheme: the number of output neurons is same as the number of classes. For each class, the first sequence is chosen to be the target of one (distinct) neuron corresponding to the class, and the second sequence is targeted for all other output neurons.

## 2.5 Datasets

### 2.5.1 MNIST

The MNIST [6] digit dataset consists of $60,000$ samples for training and $10,000$ for testing, each of which is a $28 \times 28$ grayscale image. Each pixel value of an MNIST image is converted into a real-valued input current. For the fully connected feedforward networks, the inputs are encoded from each $28 \times 28 \times N_t$ image into a 2D $784 \times N_t$ matrix where $N_t$ is the simulation time steps. Each input sample is normalized to the same mean and standard deviation. No data augmentation is applied in our experiments.

### 2.5.2 N-MNIST

The N-MNIST dataset [8] is a neuromorphic version of the MNIST dataset generated by tilting a Dynamic Version Sensor (DVS) [7] in front of static digit images on a computer monitor. The movement induced pixel intensity changes at each location are encoded as spike trains. Since the intensity can either increase or decrease, two kinds of ON- and OFF-events spike events are recorded. Due to the relative shifts of each image, an image size of $34 \times 34$ is produced. Each sample of the N-MNIST is a spatio-temporal pattern with $34 \times 34 \times 2$ spike sequences lasting for $300ms$ with the resolution of $1us$. It means there are 300000 time steps in the original N-MNIST dataset. In our experiments, we reduce the time resolution of the N-MNIST samples by 3000 times to speed up the simulation. Therefore, the preprocessed samples only have about 100 time steps. We determine that a channel has a spike at a certain time step of the preprocessed sample if there's at least one spike among the corresponding 3000 time steps of the original sample. We demonstrate the result of the preprocessed N-MNIST with 100 time steps in Table 2 of the main manuscript. Moreover, the TSSL-BP method can well train the network only with the first $90ms$ spike sequences of the original dataset which results in 30 time steps after preprocessing. The results are shown in Table 2 of the main manuscript with only 30 time steps which keep the high level of performance as well as significantly reduce the latency.

### 2.5.3 FashionMNIST

The Fashion-MNIST [11] dataset contains $28 \times 28$ grey-scale images of clothing items, meant to serve as a much more difficult drop-in replacement for the MNIST dataset. The preprocessing steps are the same as MNIST.

### 2.5.4 CIFAR-10

The CIFAR-10 [5] dataset contains $60,000$ $32 \times 32$ color images in 10 different types of objects. There are $50,000$ training images and $10,000$ testing images. The pixel intensity of each channel is converted into a real-valued input. Similar to what are commonly adopted for preprocessing, the

dataset is normalized, and random cropping and horizontal flipping are applied for data augmentation. In addition, dropout layers with a rate of 0.2 are also applied during the training of CIFAR10.