[Reviews · NeurIPS 2020]

Review 1

Summary and Contributions: The study proposed a new learning algorithm, Temporal Spike Sequence Learning Backpropagation (TSSL-BP) for synaptic optimization in Spiking Neural Networks. TSSL-BP is expected to reduce consistencies between gradient computation and loss. In addition, the authors achieved remarkable performance with short stimulus duration. The authors have compared this method to the existing learning algorithms and achieved state-of-the-art performance in standard MNIST, Fashin-MNIST and CIFAR datasets.

Strengths: Some of the existing algorithms smooth the non-differential spiking information to accommodate BP-based techniques and lead to inconsistent is gradient computation. The proposed algorithm overcomes this problem by estimating inter-neuron and intra-neuron dependencies. This method, has the ability to compute output even with short duration stimulus in contrary to existing methods that use long duration to determine the label.

Weaknesses: The formulation of net gradient computation, which is the summation of gradient estimated by inter-neuron and intra-neuron dependencies is vague and logical basis is required for such formulation. Also, the comparison of the performance on the datasets is limited to only few algorithms.

Correctness: The proposed method has provided derivation and has shown remarkable performance with short stimuli duration and training from few epochs. The methodology presented is accurate.

Clarity: The paper was well written. However, some of the sentences are repeated multiple times, especially about inter-neuron and intra-neuron dependencies.

Relation to Prior Work: The study provided brief information on previous works and its difference to the proposed method. However, some of the recent literature on BP based techniques for SNN are left out.

Reproducibility: Yes

Additional Feedback:


Review 2

Summary and Contributions: A method for training spiking neural networks through error back-propagation is presented. The model uses leaky-integrate and fire neurons and introduces a new method to approximate the gradient at the non-differentiable firing time. The model is benchmarked on simpler machine learning tasks such as MNIST and CIFAR10 and show state-of-the art performance.

Strengths: The paper reports a new method to compute gradients in spiking neural network. Although a number of other methods exist, the presented approach is interesting and works well on practical tasks. The algorithm is nicely illustrated and seems overall sound. Although the mechanism seems not very biologically plausible it may help to deepen the general understanding of spiking neural network computation.

Weaknesses: It would have been nice to see results also on more complex machine learning benchmarks like ImageNet etc. Have the authors attempted to target more ambitious benchmark data sets with their model? It is also unfortunate that all presented tasks (MNIST, CIFAR10, etc.) are not sequential in nature. This obscures the sequence learning feature of the model that is so prominent in the title. It also has never been demonstrated that spike sequences can be actually learned with this model. This raises doubts on whether sequence learning is accurately possible with this model. The learning rules are also inherently non-local. This makes them little attractive for neuromorphic hardware and also does not provide a lot of new insights to understand learning in the brain.

Correctness: The claim that "The training of SNNs is significantly more costly than that of the conventional neural networks" is not correct. Arguably the most efficient neural network implementation we know of is the brain which is (mostly) spiking. Also technological solutions like spiking neuromorphic hardware is often more efficient than standard GPU hardware. This claim should be corrected or removed.

Clarity: The paper is well structured and understandable, but it contains a number of typos and would benefit from being carefully proof-read.

Relation to Prior Work: The paper did not provide a very deep discussion of other gradient-based methods to train SNNs. Especially older work, like the Tempotron model that is very similar in nature is ignored. See: http://courses.cs.tamu.edu/rgutier/cpsc636_s10/gutig2006tempotron.pdf Also more recent work is not discussed, e.g.: https://papers.nips.cc/paper/7359-long-short-term-memory-and-learning-to-learn-in-networks-of-spiking-neurons.pdf and https://www.mitpressjournals.org/doi/full/10.1162/neco_a_01086 These should be discussed and compared to the presented method.

Reproducibility: Yes

Additional Feedback: POST FEEDBACK: The authors have further clarified. Very nice work.


Review 3

Summary and Contributions: The authors present a training method for spiking neural networks that is based on standard backpropagation. The main novelty is the way the paper deals with the non-differentiable spike non-linearity. Instead of approximating this hard-nonlinearity with a differentiable smooth function, the proposed method sidesteps this issue by relating the Post-synaptic current to the presynaptic spike times and then relating the pre-synaptic spike times to the pre-synaptic membrane potential.

Strengths: The paper reports impressive results on a number of datasets and the proposed exact BP method is a nice change from the approximate BP methods used till now.

Weaknesses: Some points that need clarification: 1)The authors argue that their approach is better because it allows neural networks to respond within very few timesteps. The results are reported after 5 timesteps. I am not sure if for such very fast respons, the dynamics of the network are playing any role in this case. For example, training in MNIST requires the correct output neuron to emit a spike in the second time step. It seems there is just one volley of activity passing instantaneously throughout the network. The network is thus more similar to binary ANNs than a SNN. The authors should comment on whether the network dynamics actually play a role in the MNIST and CIFAR10 datasets. Preferably, they should also include firing statistics with the fraction of neurons that spike during these 5 time steps, and whether any neuron manages to spike more than once. 2)L194: I do not get how du_i/dt_m is obtained by differentiating [3]. The spike time t_m is obtained by thresholding u_i, so how do you differentiate through the thresholding function? In summary, the presented BP method is interesting and, I believe, novel. The major concern is that the dynamics of the network seem to have become irrelevant since the response is obtained almost immediately. Thus a better point of comparison would be binary ANNs and not other SNNs. Minor comments: L21: “rendered energy-efficient VLSI chips..’ : the sentence is incomplete L23: “fully leveraging the theoretical computing advantages of SNNs over traditional artificial neural networks (ANNs) [14]”. I do not follow this assertion. How do liquid state machines prove a theoretical advantage of SNNs over ANNs? L28: “a par with” -> “on par with” L39: “combinations of thereof” -> “combinations thereof” L86: For completeness, write down the form of the synaptic kernel \epsilon L118 “an desired” -> “a desired” L.155 “ these approaches effectively spread each discrete firing spike continuously over time, converting one actual spike to multiple “fictitious” spikes and also generating multiple “fictitious” PSCs”: This argument is a bit hand-wavy and not backed by evidence. A smoothed spike could be tuned so that it injects the same amount of PSC into the post-synaptic neuron as a real discrete spike. Spike time is also still well-defined as the peak of the smoothed spike waveform. See [1,2] for example, where training using smoothed spikes is still able to produce networks with fine control over spike times. ----- The author comments addressed my concerns. I believe it is a good paper and thus keep my score as accept. [1]Huh, Dongsung, and Terrence J. Sejnowski. "Gradient descent for spiking neural networks." Advances in Neural Information Processing Systems. 2018. [2]Zenke, Friedemann, and Surya Ganguli. "Superspike: Supervised learning in multilayer spiking neural networks." Neural computation 30.6 (2018): 1514-1541.

Correctness: Yes

Clarity: Yes

Relation to Prior Work: Yes

Reproducibility: Yes

Additional Feedback:


Review 4

Summary and Contributions: The authors introduce a new manner of performing backpropagation in spiking neural networks, called Temporal Spike Sequence Learning Backpropagation (TSSL-BP). Their method takes both inter-neuron and intra-neuron dependencies into account and allows to learn "precise" spiking sequences on the output neurons, leading to improved learning precision on various tasks.

Strengths: - The proposed learning method seems to better take into account the discrete nature of spikes than previous methods, with attention for inter-neuron and intra-neuron effects. - The learning method leads to a high precision given a low number of necessary time steps, which is an impressive double gain.

Weaknesses: - A main remark I would have is the following. The authors state in the main article that their loss is as in Eq 6. This would mean that the error is only zero if an output fires at exactly the right time. It is 1 otherwise (or 0.5 given the 1/2 in the function). I would estimate that this setup would form a hard, needle-in-a-haystack-like learning problem, in terms of determining a sensible gradient. Indeed, in the supplementary material, the authors say that they actually use the spike response kernel in their error function (Eq 3 in SM). This kernel smooths out the spike a bit, as shown in many of the authors' figures, and gives a lower error if spikes get closer to each other, so in my eyes is likely to facilitate learning with backprop. Hence, I think that this is a very relevant part of the method. The actual formula for the error is only mentioned in the SM - but should be mentioned in the main article in my eyes. Furthermore, about the shape of that kernel, the authors only say in the main article that they adopt "a first order synaptic model as the spike response kernel." As I would guess the spike response kernel is important, I would like to see the mathematical formulation in the main article (perhaps I read over it?) and I would be interested to see how the method fares without this "smoothed" error. (Please note that it is interesting that the authors "criticize" previous methods for assuming a differentiable function for the spike because it smooths out a signal). - Currently, the results are reported for the full method. However, in order to see what the influence is of, e.g., the intra-neuron backprop part of the learning method, an ablation study would be useful. What kind of things can the method learn better when including this term? - I see no reference to e-prop: Bellec, G., Scherr, F., Hajek, E., Salaj, D., Legenstein, R., & Maass, W. (2019). Biologically inspired alternatives to backpropagation through time for learning in recurrent neural nets. arXiv preprint arXiv:1901.09049. A direct comparison with that method is I would say out of scope, but I would be interested in a reflection of the authors on the difference with that method, which relies on synaptic traces.

Correctness: As far as I can judge the methodology and claims are correct.

Clarity: The article is in general well-written.

Relation to Prior Work: Yes. As said though, I would be interested in having the authors comment on how their method relates to that of Bellec et al.

Reproducibility: Yes

Additional Feedback: In general I think this is very insightful work, with really impressive results on the given datasets. POST FEEDBACK: The authors have adequately addressed my comments. I congratulate them with their interesting work.

[Author Response · NeurIPS 2020]

We deeply appreciate all the insightful review comments. We will fix all writing glitches, improve clarity and quality of writing, correct the confusions, and cite the missing references in our final paper with the major issues responded below:

**Discussion of previous works:** We briefly comment on some references below and discuss more in our revised paper.

[1] proposes a surrogate gradient BP method called Superspike. It uses the partial derivative of the negative half of a fast sigmoid as the surrogate gradient function to circumvent the non-differentiability of spikes. In addition, the authors also investigated different feedback methods to generate error signals from the output layer to hidden layers.

[2] presents a BP method for recurrent SNNs based on a novel combination of a gate function and threshold-triggered synaptic model that are introduced to handle non-differentiability of spikes. In this work, depolarization of membrane potential within a narrow active zone below the firing threshold also induces graded postsynaptic current.

[3] proposes a new type of SNNs, Long Short-Term Memory Spiking Neural Networks (LSNNs) with adapting neurons and support for learning to learn, trained with BPTT with surrogate gradient, demonstrating very good results.

[4] factorizes the standard BPTT into a new form, and proposes three very interesting ideas of converting BPTT into more biologically plausible online learning: (1) an online method to approximate feedback errors, (2) a separate error prediction module trained in the outer loop over a family of different tasks, (3) synthetic gradients combined with eligibility traces for more accurate approximation of the error gradients.

Tempotron uses a "gradient-descent" dynamics and targets only learning timing-based decisions by *single* neurons.

We have a different focus. Superspike [1] is a BP method with surrogate gradient while we more precisely compute gradients through inter and intra dependencies at spiking times. [2] formulates BP at the level of continuous postsynaptic level without directly involving spike timing, which is our focus. In [2], if the membrane potential falls within delta below the firing threshold (activation zone), a graded post-synaptic current will be generated. Differently, we directly consider the all-or-none characteristics of firing spikes. [3] proposes a new recurrent SNN/learning-to-learn network architecture and [4] focuses on the higher-level problem of biologically-plausible online learning. In contrast, we deal with the fundamental problem of BP training with more precisely computed error gradients.

**Implementation on neurmorphic hardware:** Our TSSL-BP is not biologically plausible and may complicate the implementation on neuromorphic hardware - a limitation. It can train SNNs with high accuracy and low-latency. Low-latency would mitigate its complexity on neuromorphic hardware to a certain extent.

**Dynamics over a short time window:** We use a short time window of 5 steps to demonstrate the precision of TSSL-BP under low-latency. For most input examples, each trained SNN produces the targeted temporally-varying firing sequences at the output layer. These SNNs are not Time-To-First-Spike networks; neurons are allowed to fire multiple times. Most of the neurons either fire after the first time point or have multiple spikes. Unlike binary ANNs, the trained SNNs here are dynamical. In one SNN, about $20\%$ of neurons fire more than once, $9\%$ of neurons fire more than twice, and $4\%$ of neurons fire more than thrice. We'll include more specific firing statistics in our revised paper.

**Intra/inter-neuron dependencies:** As in 3.3.2, we split the derivative of a PSC w.r.t a presynaptic spike time $\frac{\partial a[t_k]}{\partial t_m}$ into two parts. First, the spike at $t_m$ directly affects $a[t_k]$, which is called inter dependency. Second, the spike at $t_m$ also affects the succeeding presynaptic spike $t_p$ through resetting which further affects $a[t_k]$. This secondary effect is called intra dependency. Inter-neuron dependencies are dominant in the overall gradients; including the intra-neuron part further improves performance/training speed. Including intra-dependencies in TSSL-BP boosts accuracy by $1.5\%$ for DVS Gesture dataset (40 epochs) and by $4\%$ for CIFAR10 DVS dataset (trained for 5 epochs due to time limitation).

**Kernel in loss function:** TSSL-BP is flexible about how the loss is defined. The difference between the actual output/targeted firing sequences can be defined via direct comparison, e.g. (6) in the main text, or by using a kernel to measure the so-called Van Rossum distance. The two losses lead to a small performance difference of $< 0.1\%$ for MNIST. Using a kernel to define the loss only smooths the loss but not the firing spikes in the SNN so that the problem of non-differentiable spikes still exists in BPTT with surrogate gradient. Synaptic kernel describes synaptic dynamics and is for a different purpose than the kernel used in the loss. We happen to make the two kernels identical.

**Time derivative of membrane potential:** As in (3), $\frac{\partial u_i[t_m]}{\partial t_m}$ measures the slope of the membrane potential around firing time $t_m$. $\frac{\partial u_i[t_m]}{\partial t_m}$ is computed right before the firing: $\frac{\partial u_i[t_m]}{\partial t_m} = \lim_{\Delta t \to 0} \frac{u_i[t_m] - u_i[t_m - \Delta t]}{\Delta t}$ w/o involving thresholding.

[1] Zenke, Friedemann, and Surya Ganguli. "Superspike: Supervised learning in multilayer spiking neural networks."

[2] Huh, Dongsung, and Terrence J. Sejnowski. "Gradient descent for spiking neural networks."

[3] Bellec, Guillaume, et al. "Long short-term memory and learning-to-learn in networks of spiking neurons."

[4] Bellec, Guillaume, et al. "Biologically inspired alternatives to backpropagation through time for learning in recurrent neural nets."


[Meta-Review · NeurIPS 2020]

The reviewers all agreed that this is an excellent paper that makes a clear contribution to the spiking NN literature. There were some concerns, but these were largely addressed in rebuttal.